# Insights into the Physiological and Biochemical Impacts of Salt Stress on Plant Growth and Development

**Muhammad Adnan Shahid** [1,*], **Ali Sarkhosh** [2,*], **Naeem Khan** [3], **Rashad Mukhtar Balal** [4], **Shahid Ali** [5], **Lorenzo Rossi** [6], **Celina Gómez** [7], **Neil Mattson** [8], **Wajid Nasim** [9] **and Francisco Garcia-Sanchez** [10]

1   Department of Agriculture, Nutrition and Food Systems, University of New Hampshire, Durham, NH 03824, USA
2   Horticultural Sciences Department, IFAS-University of Florida, Gainesville, FL 32611, USA
3   Department of Agronomy, IFAS-University of Florida, Gainesville, FL 32611, USA; naeemkhan@ufl.edu
4   Department of Horticulture, College of Agriculture, University of Sargodha, Sargodha 40100, Pakistan; uaf_rashad@yahoo.com
5   Plant Epigenetic and Development, Northeast Forestry University Harbin, Heilongjiang 150040, China; shahidsafi926@gmail.com
6   Horticultural Sciences Department, University of Florida, IFAS-Indian River Research and Education Center, Fort Pierce, FL 34945, USA; l.rossi@ufl.edu
7   Environmental Horticulture Department, IFAS-University of Florida, Gainesville, FL 32611, USA; cgomezv@ufl.edu
8   Horticulture Section, School of Integrative Plant Science, Cornell University, Ithaca, NY 14853, USA; nsm47@cornell.edu
9   Department of Agronomy, Faculty of Agriculture and Environmental Sciences, The Islamia University of Bahawalpur (IUB), Punjab 63100, Pakistan; wajid.nasim@iub.edu.pk
10  CEBAS-CSIC, Departamento De Nutrición Vegetal, Campus Universitario de Espinardo, Espinardo, 30100 Murcia, Spain; fgs@cebas.csic.es
*   Correspondence: muhammad.shahid@unh.edu (M.A.S.); sarkhosha@ufl.edu (A.S.)

**Abstract:** Climate change is causing soil salinization, resulting in crop losses throughout the world. The ability of plants to tolerate salt stress is determined by multiple biochemical and molecular pathways. Here we discuss physiological, biochemical, and cellular modulations in plants in response to salt stress. Knowledge of these modulations can assist in assessing salt tolerance potential and the mechanisms underlying salinity tolerance in plants. Salinity-induced cellular damage is highly correlated with generation of reactive oxygen species, ionic imbalance, osmotic damage, and reduced relative water content. Accelerated antioxidant activities and osmotic adjustment by the formation of organic and inorganic osmolytes are significant and effective salinity tolerance mechanisms for crop plants. In addition, polyamines improve salt tolerance by regulating various physiological mechanisms, including rhizogenesis, somatic embryogenesis, maintenance of cell pH, and ionic homeostasis. This research project focuses on three strategies to augment salinity tolerance capacity in agricultural crops: salinity-induced alterations in signaling pathways; signaling of phytohormones, ion channels, and biosensors; and expression of ion transporter genes in crop plants (especially in comparison to halophytes).

**Keywords:** abiotic stresses; cell membrane stability; climate change; osmolytes; polyamines

## 1. Introduction

### 1.1. Overview of Salinity

Abiotic stresses like salinity, drought, and high temperature have undesirable effects on crop productivity and quality, and negative trends in sustainable agriculture [1]. Salinity in particular is an important limiting factor, causing low yield with inferior quality. Climate change is considered one of the major contributing factors to soil salinization, leading to land degradation and desertification [2]. According to Flowers et al. [3], high salt concentration is responsible for negative impacts on 7% of total land surface, and 5% of cultivated land. Poor irrigation water quality is another important factor contributing to soil salinization [4]. For these reasons, soil salinization is a reported major cause of reductions in the productivity of irrigated and rainfed lands of the world [5,6].

The adverse effects of salinization on plants are evident from negative growth trends from alteration or inhibition of biochemical and physiological processes. Plants can be classified as glycophytes or halophytes by their ability to survive under high salt concentrations [7]. Glycophytes are plants that are severely affected by saline conditions both at the cellular and whole-plant level. Under saline conditions, these plants exhibit greater accumulation of solutes, and ionic and osmotic stresses confer nutritional imbalances, which limit the productivity of these plants. The majority of terrestrial plants are glycophytes, including crop plants [1,8].

Conversely, halophytes regulate their biochemical and physiological processes through ionic compartmentalization, production of osmolytes and compatible solutes, enzymatic changes, and absorption of selective ions. These adaptations promote seed germination, succulence, and salt exclusion for these plants in a saline environment [9,10]. Halophyte succulence keeps the proportion of ions to water in balance under high salt conditions by maintaining high water contents. Succulence is expressed as large cell size, reduced growth and surface area per tissue volume, and increased water constituents. Interestingly, halophytes also have a greater number of mitochondria, indicating that more energy is required to survive under saline conditions [11,12]. Halophytes also have less sodium and chloride ion accumulation in their cytoplasms, allowing their chloroplasts to survive even while the plant experiences salinity shocks [13,14]. Halophytes also have a specialized system for salt excretion from the plant tissues via specific glands. These glands are characteristic of halophytic leaves. The leaves will remove the salts onto the leaf surface before the salts can reach the shoots of the plant. The presence of halophytes is limited to habitats with plentiful water (e.g., salt marshes, etc.). "Salt hairs," which regulate water loss, replace "secretary glands", if a plant is adapted to a relatively drier climate as compared to marshes [15,16]. Hydathodes are another adaptation by plants to remove excessive salts, with less stomatal conductance and transpiration water loss [17,18].

Various undesirable effects appear because of high salt concentration. Ion imbalance is one of the major consequences. A high concentration of Na and Cl ions, as an example, can lead to biochemical processes which can prove to be fatal for the plants [19–21]. Sodium and chloride toxicity not only induce nutritional disorders but also cause physiological drought by lowering the osmotic potential of the soil solutions [22]. Soil salinity prevents the plant from taking up water from the soil, resulting in a decline in cellular water, thus affecting cell turgor. Soil salinity also adversely affects photosynthetic activity in the plant and encourages the production of reactive oxygen species (ROS), thus reducing plant growth [23,24].

The identification of salt stress by plant species and their subsequent response is controlled by signals—signals which are generated by ions, osmotic differential, hormones, or ROS [25]. These signals bind to their respective receptors and initiate the physiological mechanisms which enable a plant to adapt to stress conditions (Figure 1). Under abiotic stress conditions, three types of signal transduction have been categorized, i.e., the ionic signaling pathway, the osmolyte regulation pathway, and the gene regulation pathway [26]. For signal transduction under salinity stress, the ionic stress signaling pathway has been elucidated. Calcium (Ca) occupies a central position in this regard. It induces signal transduction in plants to adapt to stress conditions [27]. High cytosolic Ca concentration initiates many

processes involving enzymatic activity regulation, ion channel performance, and gene expression [28]. Exogenously applied calcium regulates K$^+$/Na$^+$ selectivity, and thus confers salt adaptation by improving signal transduction. Glycinebetaine is reported to maintain signal transduction and ion homeostasis under salt-stressed conditions [29,30].

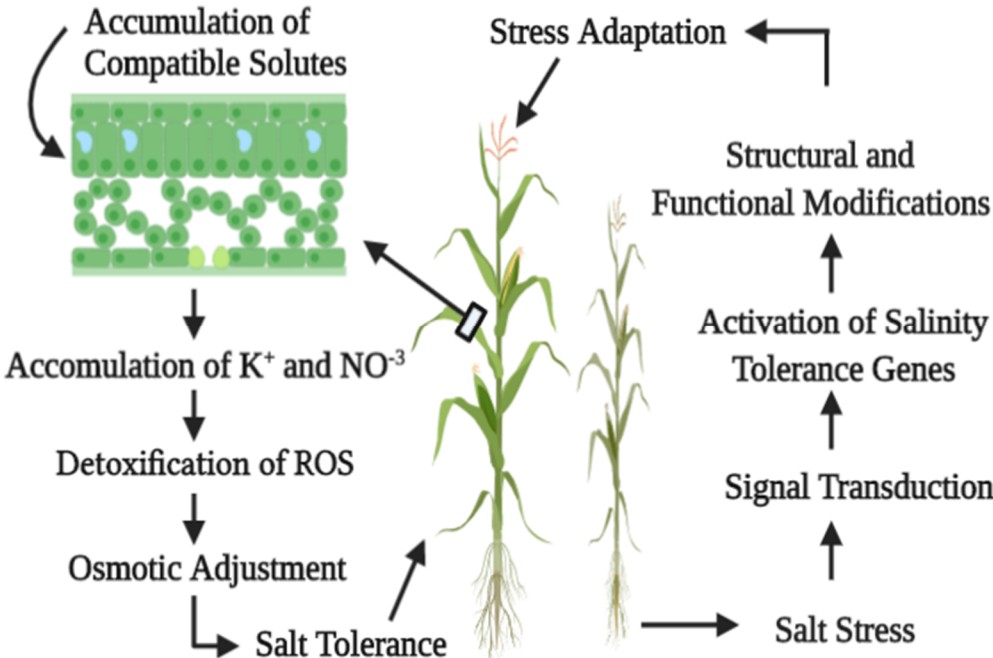

**Figure 1.** Salt stress signals that bind to their respective receptors and initiate the physiological and molecular mechanisms to enable a plant to survive under stressed conditions.

Glycophytes and halophytes mediate serious effects of salinity at the cellular level by inducing changes in the plasma membrane and cytoplasm. As a tolerance mechanism, the plant alters the structure and composition of their plasma membrane, especially lipid and protein contents. The cell membrane is usually the foremost target of any stress [31]. Salt stress also alters the cell cytoplasmic viscosity and composition [32,33].

It is essential to understand each tolerance mechanism at the cellular level in order to understand each tolerance mechanism at the plant level. The protoplasmic features studied at the cellular level include plasma membrane permeability, cytoplasmic viscosity, cytoplasmic streaming, and cell solute potential. Cytoplasmic viscosity describes the water contents of the cytoplasm in conjunction with its inter-macromolecular interactions. Cell solute potential represents the solute contents of the cell. Cell membrane permeability significantly increases with the increase in salinity [34,35]. The ability of the plasma membrane to repair, regenerate, and maintain its integrity stabilizes the cell structure and function under stress conditions. It is mainly dependent on the composition of the plasma membrane (i.e., mainly lipid contents). Saline conditions result in enhanced lipid peroxidation [36].

The cell membrane stability technique is widely utilized to judge the behavior of various plant genotypes in response to salt stress [37]. Thus, it can contribute to assessing the salt tolerance of plant genotypes. Cell membrane stability is also reported to correlate with potassium (K) ions, osmotic potential, osmotic adjustment, and relative water contents [38,39]. The differing patterns of cell membrane permeability help in characterizing genotypes as tolerant or sensitive. In a saline environment, salt-sensitive genotypes show marked alterations, whereas salt-tolerant genotypes show minor changes. Salinity also alters the degree of saturation of membrane fatty acids and membrane fluidity [40].

The salt-tolerant genotypes have high cytoplasmic viscosity due to augmentation in hydrophilic cytoplasmic proteins and other macromolecules [41]. In more sensitive genotypes, a saline environment

results in a high concentration of salts in plant cells, lowering the solute potential [42]. Salinity inflicts serious irregularities during cell division, one of the various metabolic processes which face severe alterations in a saline environment. A saline environment especially alters the leaf anatomy by affecting mitochondria and vacuoles [43,44], plant leaf area, and stomatal thickness [45]. One way plants exhibit tolerance to saline environments is portioning or compartmentalization of toxic ions. This mechanism enables salt-tolerant plant species to retain toxic levels of harmful ions in vacuoles and inhibit their interference with cytoplasmic metabolic activities [46]. The $Na^+$ and $Cl^-$ partitioning in the vacuole stimulates higher concentrations of $K^+$ and organic osmolytes in the cytoplasm in order to adjust osmotic pressure of the ions in the vacuole [47].

*1.2. Salinity and Morphological Attributes*

Halophytes have the unique feature of succulence, a feature which keeps the ionic uptake in proper proportion with water, by maintaining high water contents (Figure 2). Succulence results in large cell size, reduced growth [48,49], reduced surface area per tissue volume, and increased water constituents. The salt tolerance is evident as maintenance of vegetative growth and yield and lower necrotic percentage [50,51]. Moreover, halophytes also have a greater number of mitochondria, indicating that more energy is required to survive under saline conditions [11,21]. Sodium (Na) accumulation causes necrosis in old leaves, initiating from tips and then extending towards the leaf base. It also decreases leaf life span, net productivity, and crop yield [52]. Biomass reduction and foliar damage become more prominent with time and at higher salinity levels. High salt concentration caused a reduction in fresh fruit yield in various vegetables [31,53–55]. However, salinity treatments did not prove harmful for vegetative growth and the number of flowers [53,55]. In contrast to the above report, Chartzoulakis and Klapaki, [56], Kaya et al. [57], and Giuffrida et al. [58], noted a reduction in fruit numbers and fruit weight under salt stress. Saline conditions were reported as producing non-significant results on some growth attributes, water status, and tissue concentration of major nutrients [59]. Salt treatments caused a significant reduction in plant height, root length, and dry weight [60–62]. Salt stress caused reduction in fresh and dry weight of cotton seedlings [63,64] and seed germination percentage in wheat varieties [35,65]. Tomato (*Solanum lycopersicum*) and pepper plants grown under salt stress experienced a reduction in dry weight, plant height [57,66], fruit weight, and relative water contents [67]. Broad bean, which is a green vegetable, experienced a significant reduction in plant height, leaf area, pod weight, number of pods per plant, seed yield, number of seeds, and product quality due to salt stress. However, a significant positive trend was observed between dry leaf matter, specific leaf weight, and salinity [68]. Suppression in seedling growth and dry matter accumulation was observed in Indian mustard (*Brassica juncea* L.) because of salt stress, which was ameliorated by putrescine application [69]. Exogenously applied sugar beet extract and shikimic acid on salt stressed eggplant and tomato crops respectively displayed a marked influence on fresh fruit weight, number of fruits, and shoot and root fresh and dry weight [54,70].

According to Caines and Shennan [71], root growth is more susceptible to saline conditions than shoot growth, but both are affected, making them reasonable indicators of salinity damage. A similar suppression in the shoot and root growth was also observed by Evers et al. [72] and Gao et al. [73] in the case of *Solanum tuberosum* L. Under high salt concentrations, potato root and shoot development was hindered [74,75]. High salt concentrations reduced the leaf area and increased the root:shoot ratio in wild type (Ailsa Craig) and ABA-deficit mutant (*notabilis*) tomato genotypes [76]. Suppression of fresh and dry weight of tomato plants because of salt stress can be alleviated by promoting the growth of *Achromobacter piechaudii* in the tomato growth media. *A. piechaudii* also caused a reduction in ethylene production by tomato seedlings, the opposite effect of salt stress on tomato seedlings [77]. High ethylene levels proved to be harmful to the growth of the plant [78]. Interestingly, Hu et al. [79] reported that potato root growth could be improved through brassinosteroid application. Salt stress caused a reduction in marketable yield of pepper plants grown hydroponically. Saline conditions also imparted negative features to the fruit quality in terms of fruit pulp thickness and firmness. It also

resulted in increased fructose, glucose, and *myo*-inositol fruit concentrations [4,42]. The salt-sensitive pepper genotypes showed maximum damage and experienced severe chlorosis and necrosis, whereas tolerant genotypes were slightly less affected. Sodium exclusion can be regarded as a criterion to allocate salt stress tolerance status to pepper genotypes [80]. Korkmaz et al. [81] have reported that the exogenous application of glycine betaine can reduce the effect of salinity in pepper plants.

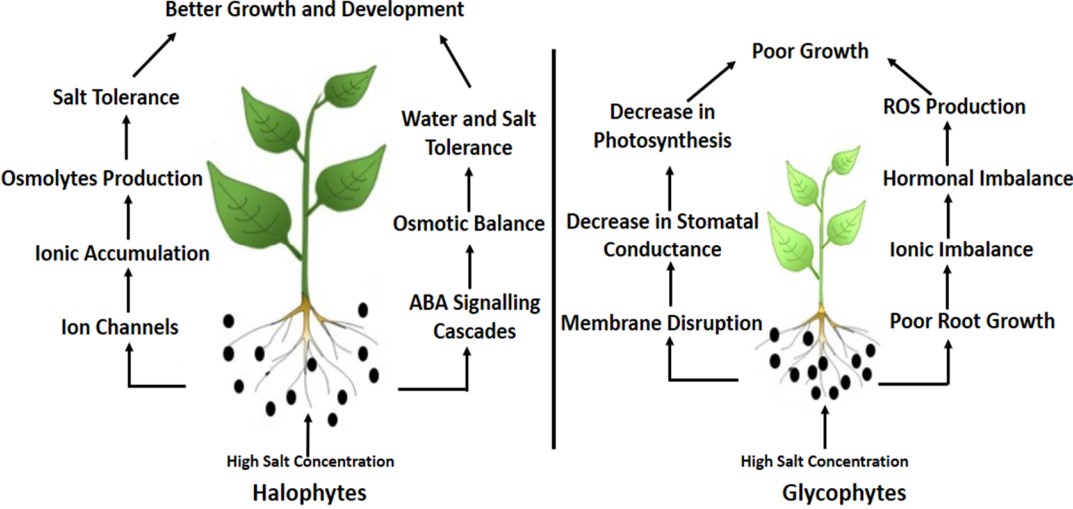

**Figure 2.** Regulation of physiological and biochemical process in halophytes through ionic compartmentalization, osmotic adjustment, enzymatic activities, polyamines, and stress signaling regulation.

High salt concentration has been observed to cause detrimental effects on leguminous crops. In fact, saline conditions induced smaller sized nodules, reduced nodule volume per plant, less nodulation, and inferior plant growth. At the cellular level, saline conditions caused drastic alteration in the mechanism of nodule formation. It reduced the turgor of the peripheral cells of the nodule, altered its zonation, enhanced the infection thread enlargement, reduced the release of bacteria from infection threads, and stimulated the electron-dense material (phenolics) and its accumulation in vacuoles. Salinity caused a reduction in nitrogen-fixing ability of the nodules, which has the outcome of reduced respiration rate and protein synthesis [82–84]. Salinity tolerance patterns vary considerably among leguminous crops. It is distinctly apparent in *Vicia faba*, *Glycine max*, *Pisum sativum*, and *Casuarina glauca*, which can be categorized as salt-sensitive [85,86]. In addition, germination of seeds faces serious limitations upon exposure to salt stress [87,88]. Halophytes and glycophytes differ significantly in their germination behavior. Halophytes have the ability to maintain their germination mechanism to an extent with the advent of salinity. However, a sharp decline in germination occurs for glycophytes under salt stress. Imbibition is affected by the lower solute potential of the soil solution. It results in enzymatic deregulations and imbalances in source-sink relationships and ratios of different plant growth regulators present in the seed and required for efficient seed germination [89,90].

### 1.3. Salinity and Physiological Attributes

Photosynthesis is of prime importance in the production of adenosine triphosphate (ATP), which provides the energy required for $CO_2$ fixation to sugars. Various abiotic stresses alter photosynthetic mechanisms [91] by disrupting thylakoid membranes, modifying the electron transport chain, altering enzymatic activity and protein synthesis, and changing Calvin cycle patterns. All these abnormalities cause deregulation in ATP synthesis [92], which can lead to the deficiency of certain ions due to ion degradation and synthesis inhibition [93–96].

Stepien and Klobus [97] reported a decline in the relative water content of cucumber (*Cucumis sativus*) leaves after exposure to saline solutions. They attributed their results to higher Na and reduced K content, a situation which reduces photosynthesis because of the antagonistic competition of Na for ion uptake. Gas exchange is inversely related to the concentration of Na and chloride ions [98,99], and parameters like photosynthesis, stomatal conductance, and transpiration tend to be negatively influenced by saline treatments [100,101]. Photosynthesis of pepper plants has been reported to be lower when the entire root system is affected by salt stress compared to partial root exposure, and factors like stomatal conductance and transpiration are similarly affected by complete or partial salt stress [99,102]. A low stomatal conductance was reported in wild type (Ailsa Craig) and ABA-deficit mutant (*notabilis*) tomato genotypes, which was negatively correlated with increasing xylem ABA for both genotypes [76]. Stomatal conductance is often correlated with photosynthetic efficiency, which is a prerequisite for higher biomass production and yield [10]. In addition, salinity is reported to reduce the maximum quantum efficiency of photosystem II (PSII) [48,94,103–106]. A distinct correlation has been found between Na ion contents and chlorophyll fluorescence, which is often used to estimate salt tolerance of plants [48]. Higher salt levels are also known to alter photosynthesis via non-stomatal limitations, including variations in photosynthetic enzyme activity and changes in the concentration of chlorophyll and carotenoids [107,108]. Pepper leaves have exhibited a significant reduction in chlorophyll pigment under saline conditions [109]. A similar reduction in chlorophyll *a* and *b* contents has been reported in melon by Kaya et al. [57]. This chlorophyll degradation under salt stress can be attributed to an enzyme called chlorophyllase [110–113] and to the absolute concentration of chloride and Na in the leaves [98,114–118]. Although carotenoid content has been reported to decline in response to salinity, anthocyanin pigments typically increase as a result of salinity.

Plant antioxidants include secondary metabolites (phenolic compounds), which are generated in response to stress conditions. These secondary metabolites may include tocopherol, which serves to stabilize membrane integrity [119,120], ascorbic acid, carotenoids, flavonoids, and glutathione [90,121]. Tocopherol plays a key role as a signaling molecule between cells [93]. Ascorbic acid is a significant antioxidant involved in plant adaptation [122] and occurs abundantly in cell organelles and apoplasts [99]. It has the ability to scavenge superoxide, hydroxyl, and singlet oxygen. Carotenoids are found in chloroplasts and reported to aid in light reception for photosynthesis. Moreover, they are also protective compounds which scavenge ROS [95,123]. Putrescine is reported to increase the level of carotenoids and glutathione in Indian mustard (*B. juncea* L.) against salt stress. Putrescine supplementation inhibits ROS generation by accelerating antioxygenic enzymes and therefore aiding in the maintenance of chloroplastic membranes and the $NADP^+$/NADPH ratio [124–126].

$C_4$ plants are reported to be more resistant against salinity stress than $C_3$ plants by having a better capacity to preserve the photosynthetic apparatus against oxidative stress [127,128]. Other physiological responses that are indirectly related to salinity stress include changes in water use efficiency and evapotranspiration, which can benefit from the use of beneficial bacteria [77,129–131]. In addition, higher salt concentration can stimulate the accumulation of spermine and spermidine, which contribute to the induction of salt tolerance in plants and lead to maintenance of fruit quality (Figure 3) [132–135].

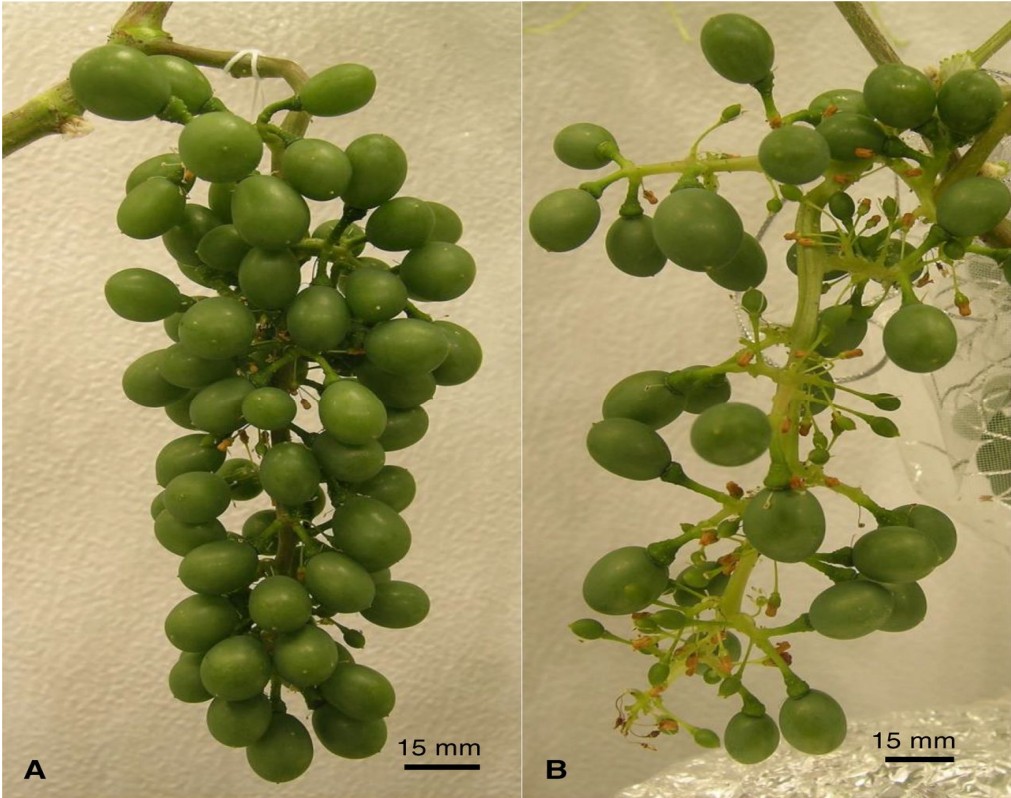

**Figure 3.** Bunches, grapevine cultivar Shiraz. Control bunch (**A**); Salt-stressed bunch (**B**); showing symptoms of coulure and millerandage. Salt treatment was applied from budburst until veraison via fertigation with 35 mM NaCl added to control nutrient solution [136].

## 2. Salinity and Water Relations

Water is an essential constituent of plant cells, supporting almost all physiological and biochemical processes contributing to plant growth and development [8]. High salt concentration hinders the movement of water from the soil to the plants by reducing the water conductivity of roots [9] and affecting relative water content at the cellular level [31,99]. Alterations in rootzone water status have profound effects in the way plants respond to higher salt concentration [1,93], but adverse effects of salinity stress can be mitigated to an extent with proper irrigation and nutrient management and by controlling the evaporation rate of plants. A recent review by van Zelm et al. [137] listed root hydraulic conductivity, osmotic potential, relative water content, leaf water potential, water-use efficiency, stomatal conductance, and transpiration, among others, as typical parameters that are commonly used to indicate water-relation responses to salinity stress, which are determinant for plant growth.

Others have shown a relationship between water potential and photosynthetic parameters. For example, a decline in $CO_2$ assimilation has been reported with the depression of water potential [126,138]. To achieve active growth, plants maintain positive turgor pressure and regulate osmotic potential. Under saline conditions plants face osmotic stress due to alterations in water potential [48,97,139]. Osmotic stress triggers cellular and whole plant responses [44,140–143], and plants cope with that osmotic stress through regulating ionic homeostasis, which induces tolerance against toxic ions and growth under unfavorable conditions [8,144]. In order to maintain osmotic adjustment, plants lower the cellular osmotic potential to help improve water uptake and adjust ionic concentration in cells [42,145]. For that purpose, osmolytes appear to be of major consideration. These may be sugars, polyols, or amino acids. Osmotic adjustment appears to be a significant and effective salinity resistant mechanism in crop plants, which can be exploited efficiently through selection and breeding efforts that target salinity tolerance in different plant species.

## 3. Salinity and Biochemical Attributes

Plants have the ability to sustain their life under a saline environment through synthesis and accumulation of compatible solutes in the cytosol. These are soluble compounds with low molecular mass. These chemical compounds can maintain physiological and biochemical processes, without having interference in these processes. The chief components of compatible solutes are sugar alcohols (mannitol, sorbitol, ononitol), quaternary ammonia compounds (glycine betaine, proline betaine), proline, and tertiary sulfonium compounds. They act to scavenge the reactive oxygen species (ROS) and inhibit lipid peroxidation, hence preventing damage at the cellular level. These compatible solutes act in favor of osmotic adjustment and prevent ROS damage at the cellular level [146,147]. These maintain macromolecular conformation in the cytosol, which may be changed due to the accumulation of charged ions, under saline conditions [8,97]. These organic compounds are termed as compatible due to their consistency with the cell's metabolism [148], and their lowering of the water potential without altering cell water contents. These organic compounds are hydrophilic in nature, having the ability to replace water present on protein surfaces [149,150], without interfering with their structure and function. These solutes play a key role in preventing the drastic effects of high ion concentration on enzymatic activities [24,151–153]. Such important roles of compatible solutes lead to osmoregulation of plant cells under osmotic stress. In addition to osmoregulation, these organic compounds have a distinct role in protein stabilization, maintenance of membrane integrity, protection of OEC of PSII from dissociation [154], and scavenging of reactive oxygen species (ROS). Mannitol, sorbitol, glycerol, proline, ononitol, and pinitol have been reported to scavenge ROS species [155].

Mannitol (sugar alcohol) metabolism in higher plants is a superior attribute, contributing to salt and osmotic stress tolerance while playing a significant role as a compatible solute. It also improves plant responses under biotic stress as well, like under pathogen infestation [156]. Mannitol is reported to be synthesized at the same time with either sucrose or raffinose saccharide. In salt-tolerant species, mannitol accumulation increases, indicating that high mannitol levels contribute to salt tolerance. Mannitol acts to scavenge the reactive oxygen species, thus protecting protein molecules [157,158]. Pinitol and ononitol have been reported to accumulate under various stresses, predominantly drought and salt stress [159,160]. Interestingly, polyols can be used as a potential biochemical marker for genetically engineered stress resistance plant genotypes [161,162].

### 3.1. Salinity and Proline

Proline is an osmolyte, an amino acid, which is thought to play a significant role in inducing tolerance in plants against stressed conditions [163]. Salt stress can result in elevation in proline levels [74]. Ethephon, when used with sodium chloride in spinach, also increased proline levels [164]. The importance of proline is highlighted by its existence in bacteria with a relationship to plants experiencing water or salinity stress. High proline levels can serve as a nitrogen source for plants during recovery [165]. The precursor of proline synthesis is glutamate, involving pyrroline carboxylic acid synthetase and pyrroline carboxylic reductase [166]. An increase was noted in the activity of pyrroline-5-carboxylate synthetase (P5CS) and a decline was recorded in proline dehydrogenase activity in potato seedlings under salt stress. These changes of enzymatic activity were more pronounced in salt-sensitive cultivars [74,167]. However, an increase in proline contents of potato clones was recorded upon salt exposure [50,168]. It serves to stabilize ultra-structural changes in cells, scavenge ROS (reactive oxygen species), and maintain cellular redox potential. Under stress conditions, a higher accumulation of proline is reported in cell cytosol, strengthening the ability of the cell to make ionic adjustments. Its accumulation is linearly related to stress tolerance in plants [169]. Proline biosynthesis is reported to be mediated by Ca [170–172] and abscisic acid [8]. Previously, contrasting views about proline accumulation were also reported in plants under stress [19,158], where it appeared as a salt stress injury symptom, e.g., rice [173] and sorghum [174].

Some plant genotypes do not respond to proline accumulation, but their salt tolerance potential can be enhanced through the exogenous application of proline [31,175]. It may be helpful in counteracting

the harmful effects through osmo-protection, resulting in a higher growth rate. Proline also increases the activities of antioxidant enzymes like SOD (superoxide dismutase) and POD (peroxidase) [176]. Proline is not reported to scavenge ROS directly, but through enhanced antioxidant enzyme activity. It is reported to be more effective in mitigating the drastic effects of salinity than glycine betaine [177]. Proline used at higher concentrations may prove to be lethal for the plant, causing ultra-structural damages leading to ROS generation [178]. The effective dose of proline varies with genotype and plant developmental stage [179–182]. Proline accumulation has been reported for drought sensitive and tolerant barley genotypes grown under saline conditions. Under salt stress, a considerable amount of proline was present, with relatively lower quantities in root tissues. Proline accumulation is reported to be more prominent in tolerant genotypes [183].

### 3.2. Salinity and Polyamines

Polyamines are multivalent compounds consisting of two or more amino groups. In higher plants, these are identified as putrescine, spermidine, and spermine [147,184]. These are involved in various physiological mechanisms including rhizogenesis, somatic embryogenesis, maintenance of cell pH and ionic balance [29], pollen and flower formation, abscission, senescence, and dormancy. Endogenous polyamine synthesis can be stimulated by cytokinin [185,186]. These compounds act to stabilize macromolecules like DNA and RNA. Moreover, polyamines have a significant role in numerous abiotic and biotic stresses [151,187]. At the cellular level, polyamines contribute to regulating the plasma membrane potential, ionic homeostasis, and tolerance against salinity [188,189]. Exogenously applied polyamine or ornithine caused a reduction in proline accumulation in plant tissues under salt stress. However, an alternate trend was observed in the case of non-stressed beans [99,190]. Putrescine is characterized as a de-stressor agent and a nitrogen source under stressed conditions [191]. Putrescine has been reported to reverse the biomass reduction in Indian mustard [68,192]. Its production in plant cells follows two alternative pathways: conversion from ornithine or arginine. Putrescine is then converted to spermidine and subsequently to spermine by addition of an aminopropyl group. Spermine deficiency caused Ca ion imbalance in *Arabidopsis thaliana*, thus indicating spermine as a maintainer of plant cell ionic homeostasis under salt stress [193].

### 3.3. Salinity and Glycine-Betaine

Glycine-betaine (GB) is present in a wide range of organisms, from bacteria to higher plants and animals. In addition to being involved in osmoregulation, it maintains and regulates the performance of PSII protein complexes by protecting extrinsic regulatory protein against denaturation. It also stabilizes macromolecules, due to its ability to form strong bonds with water [136]. It protects these macromolecules during drought and thermal stress, which is why it is sometimes called an "osmoprotectant" [129,194]. Glycine-betaine accumulates in some crops under stress, like members of family *Poaceae* and *Chenopodiaceae* [195], and is absent entirely from other plants, like rice and tobacco. This directed the scientists to develop transgenic plants that have the ability to produce GB. In transgenic plants, the reproductive organs are capable of tolerating abiotic stresses if they can accumulate GB [195]. The precursor for GB is choline, and the conversion is managed by enzymes like choline monooxygenase and betaine-aldehyde dehydrogenase [151,196]. Choline supplementation to the growth media of the salt-stressed plant can act to restore the suppressed growth [197]. GB is water soluble, is not harmful at higher concentrations, and accumulates mainly in plastids and chloroplasts. Exogenous application of GB promotes salinity tolerance in plant species which do not naturally produce GB. A plant can utilize exogenously applied GB via leaves [198], as well as roots [199]. After absorption, GB is translocated in phloem [99,200]. GB is not directly involved in scavenging ROS species, but it alleviates the damaging effects of ROS by promoting enzymes responsible for the destruction or production suppression of ROS [201].

The reproductive stage of any plant during GB application is considered critical to ensure maximum yield. In various studies, it was reported that the plant reproductive organs acquire higher levels of

GB than the vegetative parts under stressed conditions. This indicates that high GB accumulation is more necessary for protecting the reproductive organs than it is for protecting the vegetative tissues from abiotic stresses, indicating that application timing is key [181,202]. The natural GB accumulating species include sugar beets, spinach, wheat, barley, and sorghum. High GB concentration is linearly linked with increased tolerance. Osmotic adjustment is the major mechanism involved in increased tolerance to abiotic stresses, especially salt stress. GB is responsible for turgor maintenance through osmotic adjustment [54,182,203]. However, this relationship is not satisfactory in some cases like *Triticum* spp. and *Agropyron* spp. [204]. Thus, this relationship varies with genotype [158]. The plant species which do not produce GB naturally can give a satisfactory yield and survival rate under salt stress conditions through the exogenous application of GB [42,205]. Exogenous GB, once applied, is transported rapidly throughout the plant. Exogenous application of GB has been reported in many plant species, including tobacco, rice, soybean, barley, and wheat. In barley, GB application improved stress tolerance by lowering water potential, which improved survivability. GB plays a role in osmotic adjustment and ionic homeostasis by maintaining high $K^+$ concentration compared to $Na^+$ ions. Exogenous application of GB also increased the $K^+/Na^+$ ratio [206–208]. GB also protects the photosynthetic apparatus. It enhances photosynthetic activity through increased stomatal conductance and reduced photorespiration [42,209,210].

In contrast to its positive influence, some researchers have also suggested neutral or somewhat negative responses to exogenously applied GB in some plant genotypes. For example, it appeared to have a neutral influence on growth in cotton [211], turnip, rapeseed, and tomato [212]. For the commercial application of GB, the rate, duration, timing, and frequency should be considered [158,213]. It can be used for seed treatment as well as foliar application. The application method is dependent on the plant material on which it will be applied, the timing of the application relative to plant developmental stage, and environmental conditions during the time of application [214].

Exogenously applied GB improved salt tolerance in rice by improving relative water contents in the leaves and increasing antioxidant levels, including superoxide dismutase, ascorbate peroxidase, catalase, and glutathione reductase (GR) [213]. Reduction in peroxidase activity was reported in a salt-tolerant rice genotype under salt stress. GB is also reported to reduce lipid peroxidation [215–217]. GB can prevent membrane adulterations due to osmotic stress more efficiently than proline [218]. Proline accumulation in leaves of salt-stressed plants is not reported to be correlated with exogenously applied glycine betaine [54,180]. Sugar beet is identified as the foremost source of GB [158,219]. It is appreciated as a valuable source of GB along with other beneficial compounds and is useful in inducing tolerance against salt stress in eggplant (*Solanum melongena* L.) as compared to pure GB. It has a marked influence on the morphological (growth and yield) as well as physiological and biochemical (gas exchange, photosynthetic rate, transpiration, GB accumulation) attributes [186,220].

## 4. Salinity and Enzymatic Attributes

Generation of reactive oxygen species (ROS) like singlet oxygen, superoxide radical, hydrogen peroxide, and hydroxyl radical as a consequence of exposure to various abiotic stresses causes injury to plants. Molecular oxygen is non-reactive and requires electron donors for the production of reactive oxygen species. These electron donors are typically metal ions [24,221]. Increased production of ROS is an indicator of plant stress [222] for proteins, lipids, pigments, DNA, and other molecules at the cellular level [223,224]. Plants combat ROS through enzymatic and non-enzymatic mechanisms. The enzymes used for scavenging ROS include superoxide dismutase (SOD), catalase (CAT), peroxidase (POD), ascorbate peroxidase (APX), glutathione reductase (GR), and glutathione-synthesizing enzymes [197,205,225]. However, increased generation of ROS scavenging enzymes does not always correspond to higher salinity tolerance. Many factors contribute to the effectiveness of antioxidant systems, including the site of antioxygenic enzyme production, enzyme action, and interaction of different antioxidant enzymes, according to Blokhina et al. [226]. Das and Roychoudhury [227] and Sairam et al. [228] reported increased SOD, APX, CAT, and GR activities in intolerant wheat plants

under abiotic stresses. The site of superoxide radical synthesis is reported to be chloroplasts [229,230], mitochondria [231], and microbodies. Ascorbate peroxidase (APX) is recognized as a reducing agent for $H_2O_2$ to water by the utilization of ascorbate and release of monodehydroascorbate (MDHA). MDHA reductase contributes to the conversion of MDHA to AsA (ascorbate). Ascorbate peroxidase causes effective regeneration of AsA and interrupts the cascade of oxidation caused by $H_2O_2$ [24,232]. The antioxidant response of plants can be regulated by compounds like $H_2O_2$ under stress conditions. $H_2O_2$ contributes almost 50% of the destruction caused by oxygen radicals in photosynthetic reduction. In terms of negative impacts on plant physiology, $H_2O_2$ is the most harmful of all reactive oxygen radicals. APX exists in four different forms: chloroplast stromal soluble form (sAPX), chloroplast thylakoid bound form (tAPX), cytosolic form (cAPX), and glyoxysome membrane form (gmAPX) (Figure 4). $H_2O_2$ increased the level of other antioxidants and caused a decline in lipid peroxidation in maize under stressed conditions [210,211,233]. Saline conditions applied to potato seedlings and *Broussonetia papyrifera* showed an increase in APX activity, new POD and SOD isoenzyme activities, and alterations in isoenzyme composition [74,212]. Similarly, seed treatment of wheat by $H_2O$ enhanced the salinity tolerance of young seedlings of wheat through prevention of oxidation damage and induction of stress proteins [234]. Exogenously applied GB has been reported to enhance antioxidant activity in terms of SOD, ascorbate peroxidase, CAT, and GR in salt-tolerant rice and wheat genotypes under salt stress conditions [197,214,235].

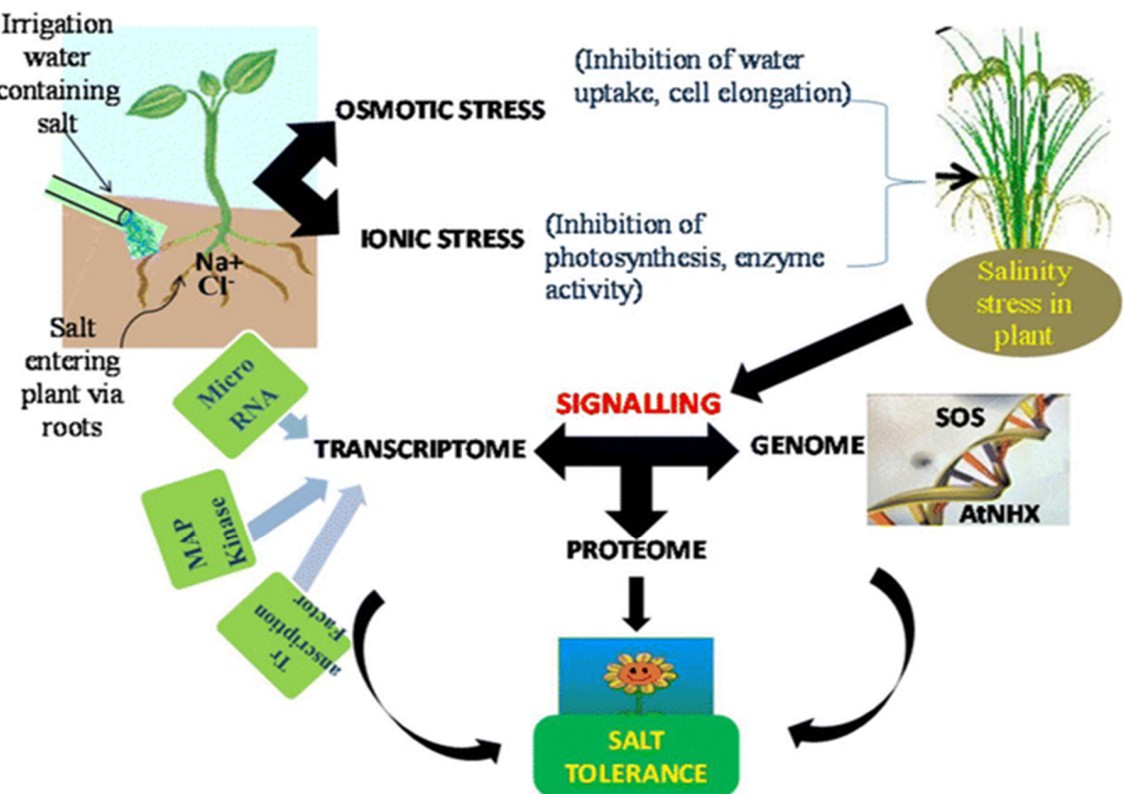

**Figure 4.** Schematic diagram showing routes of salt stress toxicity and various tolerance strategies in plants. Putative roles and action of genes, transcription factors, mitogen-activated protein kinases, microRNAs, and metabolites are shown [236].

Superoxide dismutase (SOD) activity is reported in almost all plant cell types. It causes disproportionation of singlet oxygen to molecular oxygen and hydrogen peroxide. The isoforms of SOD are categorized as copper- and zinc-containing superoxide dismutase (Cu/Zn-SOD), manganese-containing superoxide dismutase (Mn-SOD), and nickel- and iron-containing superoxide dismutase as Ni-SOD and Fe-SOD [79]. A negative correlation was observed between proline

accumulation and SOD activity in wild halophytes, so SOD activity does not necessarily induce salt tolerance in plants [216,237]. Salt treatments caused a reduction in SOD activity in potato cultivars [74]. Catalases (CATs) contribute to the plant defense system, and are synthesized in peroxisomes and glyoxysomes. CAT is responsible for the conversion of hydrogen peroxide into water and oxygen. Under saline conditions, SOD activity was reported to be more pronounced in $C_3$ (wheat) as compared to $C_4$ (maize). Both types of plants showed the same level of elevation in APX. However, an increase in GR level was more prominent in maize plants [118,238].

Exogenous application of potassium nitrate alleviated salinity effects in winter wheat by promoting some antioxidant enzymes [60,239]. Similarly, coronatine, which has properties similar to methyl jasmonate, is reported to promote the activities of antioxidant enzymes including superoxide dismutase (SOD), catalase (CAT), peroxidase (POD), and glutathione reductase (GR), and DPPH (1,1-diphenyl-2-picrylhydrazyl) scavenging behavior in cotton leaves. DPPH-radical scavenging determines non-enzymatic antioxidant activity. COR causes a decline in generation of reactive oxygen species by increasing the activity of antioxidant enzymes [240–243]. COR is functionally similar to jasmonic acid, which mitigates salt stress [244–246]. Polyamines applied to salt-stressed plants can act to ameliorate the negative impact of salinity (i.e., growth suppression) in many crop plants [247].

## 5. Salinity and Phytohormones

Plant hormones are signaling molecules with the ability to alter the physiological mechanisms of the plant, even if present in very minute quantities [248]. Common plant hormones are auxins, gibberellins, cytokinins, abscisic acid, and ethylene [249]. Plants growing under salt stress experience imbalances in hormonal homeostasis. Stressed conditions drastically alter physiological mechanisms of the plant, creating massive changes in endogenous hormonal contents. Higher concentrations of toxic ions are negatively correlated with the levels of plant hormones like gibberellins, auxins, and cytokinin and are positively associated with the abscisic acid level [250,251]. Exogenous plant hormone application on salt stressed plants was found to alleviate the negative effects of salinity on the morphological (leaf area, dry mass), physiological (chlorophyll content, stomatal conductance, photosynthetic rate), and yield characteristics of crops [252,253].

Abscisic acid (ABA) and ethylene are involved in signaling under stress conditions. An increase in ABA concentration in plant cells has been reported under saline conditions [254]. Carotenoids are the precursor for ABA synthesis, with roots and leaves the sites of synthesis [255]. Water deficit in the root zone causes ABA generation in roots, and ABA transport to shoots is xylem mediated. The increase in pH of xylem sap increases the transport of ABA to the guard cells, where it regulates the stomatal opening and closing through the involvement of Ca ions [256]. Alterations in ABA levels indirectly affect photosynthesis through disruption in stomatal opening and closing. The photosynthetic efficiency of the plant cell declines along with deregulation of translocation and assimilate partitioning of photosynthates [8,257].

Exogenously applied plant growth regulators have been widely reported to enhance stress tolerance in numerous plant species [258]. ABA is reported to be useful in alleviating plant salt stress under low water potential. ABA production in the plant cell is also related to ethylene synthesis under salt stress. The interaction between these two stress hormones is apparent from vegetative growth and seed germination under salt stress. During root inhibition by salt stress, ethylene regulates the ABA concentration. However, the reverse has been reported in the case of seed germination [259]. Salicylic acid has displayed a defensive role in plants experiencing stress, signaling the plant to adapt to the stressful environment [260].

*Salinity and Growth Regulation*

Brassinosteroids (BRs) are growth regulators which mitigate adverse growth patterns caused by salinity. It improves the germination of seeds experiencing salt stress. The improved germination rate has been reported for rice [261] and tobacco. Application of BRs as a seed treatment enhanced the growth of rice seedlings under salt stress [262]. It helps the plant to retain its green pigments and enhances nitrate reductase activity [263–265] and nitrogen-fixing capability. Brassinosteroids (28-homoBL) increased dry matter accumulation and seed yield [266]. Foliar application of brassinosteroids (24-epibrassinolide) on pepper plants grown with saline water greatly affected shoot growth parameters and leaf water contents as compared to roots. However, its effect on chlorophyll fluorescence was non-significant [267,268]. Similar patterns of brassinosteroidal effects were observed in wheat grown under salt stress. 24-epibrassinolide application on salt-stressed wheat seedlings exhibited non-significant results in terms of plant biomass, chlorophyll content, photosynthesis rate, substomatal $CO_2$ concentration, and water use efficiency. The incremented water use efficiency can be related to higher transpiration rate shown by salt-stressed wheat seedlings, as a result of 24-epibrassinolide application [269,270]. The efficiency of exogenously applied brassinosteroids to mitigate salinity effects varies with plant species, appropriate growth stage, dose, frequency, and method of brassinosteroidal application [271–273]. The results of BR application also vary with climatic conditions—mainly temperature, light duration, and applied fertilizers [274]. Brassinolide application to salt stressed *Vigna radiata* caused enhancement in growth, photosynthetic rate, and maximum quantum yield of PSII. Generally, brassinolide has the potential to protect the photosynthetic apparatus under salt stress. It also contributed to improving the membrane stability index and leaf water potential. However, no significant results were recorded in the case of lipid peroxidation and electrolyte leakage. Brassinolide increases antioxidant enzyme and proline contents [275–278]. Increases in proline level create a protective shield when the plant is under stress by acting as a source of carbon and nitrogen, a stabilizer of the plasma membrane, and an oxygen radical scavenger [279]. Brassinosteroids also increase pigment levels in the plant [280]. BRs also improve the nitrate and nitrite reductase activity in *Vigna radiata* under salt stress. This effect can be attributed to the ability of BRs to modulate transcription and translation at the gene level, and to increase cell nitrate uptake. Increased stress tolerance caused by brassinolide application is observable as improved growth parameters such as shoot length, root length, and plant biomass [106,281].

## 6. Salinity and Carbohydrate (Sugars) Metabolism in Plants

Salinity stress progressively depletes carbohydrates in plant leaves and roots [282]. Young leaves have more hexose and starch accumulation compared to older leaves. Carbon metabolism can be used to assess the salt tolerance of a plant species. Salt-tolerant genotypes accumulated more sucrose than salt-sensitive genotypes. The role of carbohydrates in osmotic adjustment has also been authenticated [231,283]. Previous studies have reported a significant correlation between sugar accumulation levels and stress tolerance in various plant species [284]. This augmentation in sugar contents may be related to the high rate of sugar hydrolysis via hydrolytic enzymes [127,285]. Sugars stabilize the plasma membrane during plant stress by interacting with phospholipids. Under salt stress, a higher accumulation of carbohydrates in leaves may contribute to osmotic adjustment [122,286]. The higher chloride contents can cause increases in carbohydrate levels in plant tissues or starch degradation in sensitive species, whereas salt-tolerant species have less starch accumulation [287]. The carbohydrate accumulation rate varies among salt-tolerant species [147,288].

## 7. Salinity and Root Apoplastic Barriers

Plant roots are the main organ involved in the uptake of water and nutrients from a solution, whereas xylem vessels allow nutrient transport to the aerial tissues. In addition, plant roots function as the primary site for sensing salinity levels so that the plant can respond rapidly to maintain functionality. Roots can exclude and/or counteract potentially harmful substances by modifying their anatomy [289]. In particular, the endodermis that separates the cortex from the central cylinder is characterized by the development of specific wall modifications, called "apoplastic barriers" [290], formed by a combination of Casparian strips and suberin lamellae. A significant anatomical change in the root system due to salt stress is the deposition of hydrophobic polymers such as cutin and suberin on the cell wall, polymers that are often associated with hydrophobic compounds (e.g., waxes). Rossi et al. [291] reported that different apoplastic adjustments in roots modify $Na^+$ fluxes to the shoots of olive trees exposed to up to 120 mM NaCl. Similarly, Krishnamurthy et al. [292] showed that the $Na^+$ bypass flow in rice roots was reduced by the deposition of apoplastic barriers. These findings substantiated the role of root apoplastic barriers in plants' tolerance to salt stress. Afterwards, several studies have confirmed the formation of plant root apoplastic barriers as a response to different environmental stresses such as heavy metals [268], salt, and drought stress [291]. Overall, the literature indicates that plants react to environmental constraints by developing apoplastic barriers close to the root apex to mitigate the intrusion of toxic ions. This is a specific anatomical response by roots when exposed to hostile environments.

## 8. Salinity and Ionic Attributes

Abiotic stress alters the patterns of nutrient availability and transport, thus causing enormous changes in plant growth. $Na^+$ and $Cl^-$ ions are the chief competitors which restrict nutrients like $Ca^{2+}$, $K^+$, and $NO_3^-$. The increase in uptake of $Na^+$ and $Cl^-$ was more prominent in mature leaves compared to young actively growing leaves [59,293]. Plants experience a deficiency of both macro and micronutrients under salt stress. However, the extent to which salinity affects micronutrient availability is determined by plant type, growing conditions, and nutrient concentration [294,295]. Hence an effective fertilization regime in salt-affected areas can be utilized to overcome the negative effects of salinity [296,297]. The increase in nutrient availability to plants enhances the plant's ability to survive under stressed conditions [1].

### 8.1. Salinity and Ionic Homeostasis

Salinity impairs the ionic balance of the cells. Maintaining ionic balance in plants is important for increasing plant survivability under salt stress. Plants maintain ionic balance by various processes, including cellular uptake, sequestration, and ion inclusion and exclusion [8,298]. Early maturing clones of diploid potato accumulated Na ions in their leaves while late maturing varieties excluded Na ions from their leaves (Figure 5). Late maturing varieties of potato were more salt-tolerant compared to early cultivars. The lower leaves of the potato plant accumulated more Na ions than the higher leaves in both the tested potato cultivars. Late maturing genotypes also established higher K to Na ratio, conferring salt tolerance to them [50,298]. Saline conditions alter the ionic concentration of soil solutions, with an increase in $Na^+$ [74] and $Cl^-$ ions, particularly [54,299]. With the increased absorption rate of Na and chloride ions, a significant decline in other ions (e.g., K, Ca, and Mg) occurs [54,297,300].

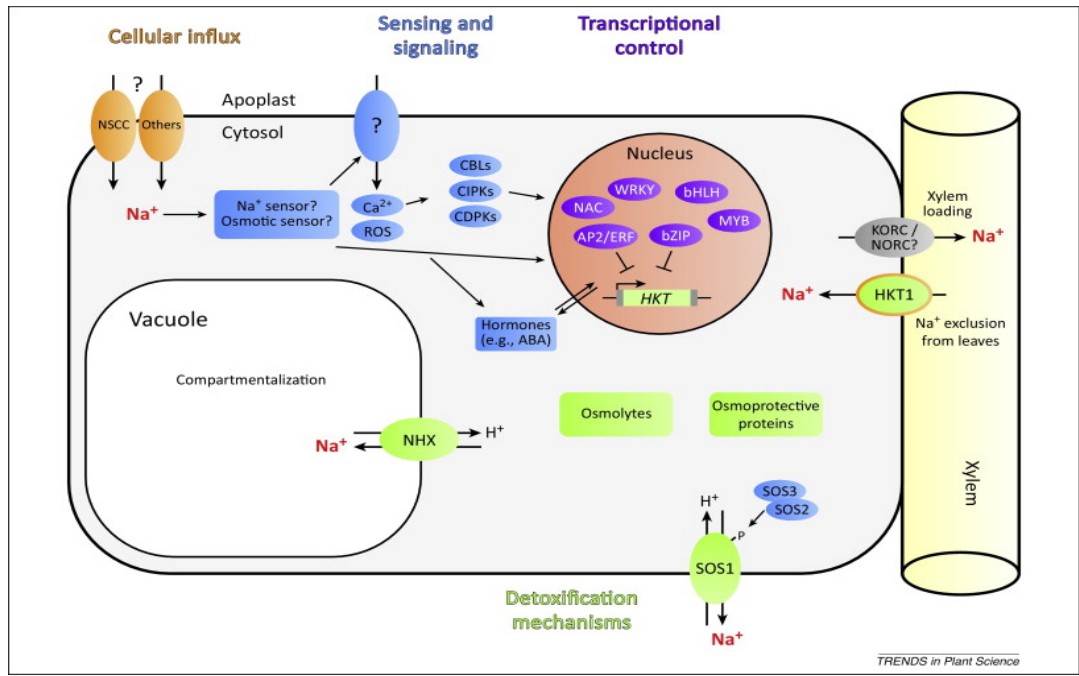

**Figure 5.** Overview of cellular Na$^+$ transport mechanisms and the salt stress response network in plant root cells. Na$^+$ (depicted in red) enters the cell via NSCCs and other membrane transporters (cellular Na$^+$-influx mechanisms highlighted in orange). Inside the cell, Na$^+$ is identified. This activates Ca$^{2+}$, ROS, and hormone signaling cascades. CBLs, CIPKs, and CDPKs are part of the Ca$^{2+}$-signaling pathway (sensing and signaling components highlighted in blue), which can alter the global transcriptional profile of the plant (transcription factor families in the nucleus depicted in purple; an AP2/ERF and a bZIP transcription factor that negatively regulate *HKT* gene expression are shown as an example). Ultimately, these early signaling pathways result in expression and activation of cellular detoxification mechanisms, including HKT, NHX, and the SOS Na$^+$ transport mechanisms, as well as osmotic protection strategies (cellular detoxification mechanisms highlighted in light green). Furthermore, the Na$^+$ distribution in the plant is regulated in a tissue-specific manner by unloading of Na$^+$ from the xylem. Abbreviations: NSCCs, nonselective cation channels; ROS, reactive oxygen species; CDPKs, calcium-dependent protein kinases; CBLs, calcineurin B-like proteins; CIPKs, CBL-interacting protein kinases; AP2/ERF, APETALA2/ETHYLENE RESPONSE FACTOR; bZIP, basic leucine zipper; NHX, Na$^+$/H$^+$ exchanger; SOS, salt overly sensitive [301].

Sodium is not considered an essential element for plant growth. Plants do not show any Na requirements, and they lack any particular transport system for Na. However, when the plant is exposed to high Na concentrations, Na finds its way into the plant cells following various pathways. It may gain entry into plant cell passively from soil solution osmolarity, or via voltage-dependent and independent cation channels [8]. Due to the similarity of hydrated ionic radii for Na$^+$ and K$^+$, Na$^+$ hampers K$^+$ absorption by the plant because transport proteins cannot distinguish between them. At higher a cystolic Na$^+$/K$^+$ concentration ratio, cellular processes are affected, which would otherwise be maintained by K$^+$, such as protein synthesis and enzyme activity [302]. Reduced K uptake can cause reductions in plant growth and productivity under saline conditions. Plants restrict the free cytosolic movement of Na in the cytosol by vacuole compartmentalization, in order to prevent Na from hampering the regular functioning of cytosolic enzymes. This mechanism is equally important for both glycophytes and halophytes [303]. Salt tolerance of a plant species is dependent on its ability to restrict translocation of toxic ions in shoots [304,305]. This ability is regulated by specific tissues [17], morphological features [306–308], and water use efficiency. These adaptive mechanisms alter the plant response to salinity at both cellular and entire plant levels [17,285,309].

## 8.2. Ionic Influx and Efflux

Sodium and chloride ions are the major ions which induce harmful effects on plant growth. The accumulation of these ions in leaves and roots determines the tolerance of a genotype to salt stress [310]. Roots are the primary organ which permits the entry of toxic ions into the plant under high saline conditions. Plant roots regulate Na and chloride contents via the extrusion mechanism. Plants respond to toxic salt levels either by exclusion to soil or by upward movement of ions through the xylem transpiration stream [311]. Phloem is a chief pathway for toxic ion transport from shoots to roots [17,278]. A passive mode of Na transport from external medium into the cell involves uniporters or ion channel type transporters like HKT, LCT1, and NSCC [45,312]. In roots, Na reaches the xylem using symplastic and apoplastic routes from the epidermis. The rate of Na transport is dependent on the barriers posed by the casparian strip and the Na extrusion from the cell. Sodium ion transport is also mediated by high-affinity potassium transporters which act as $Na^+/K^+$ symporters and also as $Na^+$-selective uniporters [313].

High salt concentration causes an increase in electrolyte leakage [106]. Sodium ion accumulation at the cellular level is a major consequence of saline environmental conditions. When Na ion levels are high, Na ions are either extruded or compartmentalized in vacuoles via $Na^+/H^+$ antiporters [45,314]. Ion transporters like $Na^+/H^+$ antiporters are involved in $Na^+$ ion extrusion or $Na^+$ ion compartmentalization in vacuoles. These antiporters may be plasma membrane-localized or tonoplast antiporters, which utilize the pH gradient, developed by P-type $H^+$-ATPases, respectively [7,8]. Electrochemical potential channels are the other source of ion transport across the cell membrane. These channels are ion specific and regulate ionic movement through gating, the period during which channels are open or closed [315].

Sodium extrusion and vacuolar compartmentalization are active (energy requiring) processes compared to sodium influx (passive transport). $K^+$ transporters are considered responsible for sodium influx into the cell. Movement of $Na^+$ ions from the cytosol to the vacuole is accomplished by $Na^+/H^+$ antiporters [316]. The plant's capacity to tolerate saline conditions depends on a high $K^+/Na^+$ cytosolic concentration. Plants adopt various strategies to cope with a high Na ion concentration, such as the restriction of $Na^+$ entry into the cell, $Na^+$ extrusion, and vacuole compartmentalization [279]. Sodium compartmentalized in the vacuole is an osmolytic process which provides a mechanism for water uptake. Sodium efflux is considered an adaptation of salt-sensitive plants, whereas Na compartmentalization is considered a feature of salt-tolerant species. The high salinity tolerance of halophytes can also be attributed to their ability to confine Na ions to their roots, making them Na accumulators [45,317]. Sodium exclusion from the plant body may also involve salt glands, present on the leaf surface [159].

Chloride also contributes to the undesirable effects of salt stress on plant growth. Salt-tolerant plant genotypes have the ability to inhibit chloride uptake. Chloride uptake is dependent on the shoot to root ratio and follows a passive transport system [286,318]. Higher chloride contents also induce succulence in salt-stressed plants [298], as well as a reduction in nitrate reductase activity [319]. Nitrate reductase activity inhibition is an indicator of poor nitrogen assimilation, leading to reduced protein synthesis and plant growth [320].

Adjusting the osmotic potential of the cell by K accumulation in vacuoles through different K channels and transporters appears to be a prominent plant strategy to cope with plant stress [321,322]. Potassium alters the membrane potential and turgor, maintains enzyme activities, and adjusts osmotic pressure and stomatal movement. It aids the plant in photosynthesis, protein synthesis, and oxidant metabolism [323–325].

## 9. Effects of Salinity on Potassium and Calcium

Potassium regulates protein synthesis, enzymatic metabolism, and photosynthesis. Low K levels were recorded in potato seedlings experiencing salt stress [31,73,326]. Potassium competes with Na under saline conditions. In saline soil, Na ions which reach the plasma membrane of a cell cause the

membrane to depolarize and the K outward rectifier channels to open. This results in a loss of K from the cell [327]. Importantly, the $K^+:Na^+$ ratio determines the saline resistance of plants. Higher $K^+$ concentration is directly related to higher biomass production. Salt-tolerant plant species have the ability to retain a higher concentration of K [72,159].

Potassium and Ca occupy a critical position in the regulation of cell membrane integrity and function [328]. Calcium aids in various physiological processes, including solute movement, stomatal regulation, molecular signaling for cell defense systems, and cell repair under stress. Under saline conditions, $Na^+/Ca^{2+}$ interactions are important because $Na^+$ has the ability to displace $Ca^{2+}$ from its binding sites, thus causing a decline in $Ca^{2+}$ availability. The cytosolic Ca concentration determines the salt sensitivity of the plant [8,329]. Calcium deficiency is evident in plants grown in saline soil conditions [330]. Calcium supplementation is beneficial in ameliorating the negative effects of salinity in beans [331,332]. The ability of plants to survive under osmotic stress is dependent on the plant's capacity to maintain high $Ca^{2+}:Na^+$ ratio and to exclude $Na^+$. Increasing $Ca^{+2}$ (by addition of $Ca^{2+}$ as gypsum $CaSO_4$) had an antagonistic effect on $Mg^{2+}$ availability to the plant, as it removed $Mg^{2+}$ from the soil complex [72,333].

Salinization management through $Ca^{+2}$ supplementation is reported to be beneficial in enhancing the quality of celery by reducing the incidence of black heart. Calcium supplementation in the form of calcium sulfate improved the growth of tomato plants. This stabilized membrane permeability, increasing N and K concentrations in leaves [67]. Calcium sulfate has more pronounced effects on tomato growth parameters than calcium chloride, perhaps because calcium chloride can be a source of chloride ions [188]. The supplementation of K and $Ca^{2+}$ to the salt-affected pepper plants increased vegetative growth and fruit production and decreased the incidence of blossom end rot, but caused a reduction in fresh fruit weight and marketable yield [4]. Calcium acetate positively stimulates photosynthesis and stomatal conductance. It also ameliorates the effects of low water content on osmotic potential from salt stress. However, $Ca^{2+}$ is reported to have an inhibitory effect on proline accumulation, which affects the low osmotic potential [34,334].

*Salinity and Nitrogen*

Nitrogen is required in considerable quantities to satisfy the mineral needs of the plant. Moreover, nitrogen is a constituent of amino acids and nucleic acids [311]. Salt stress decreased protein contents in potatoes [335]. Nitrogen has a large effect on plant growth when there is ample water. Still, addition of nitrogen to plants, even under salt stress conditions, improves the yield in many crop plants, including tomato, millet, and wheat. Nevertheless, in a saline environment, plants have a decreased ability to uptake nitrogen [1,31]. Plants that are more susceptible to $Cl^-$ toxicity can be managed through $NO_3^-$ application since $NO_3^-$ antagonizes $Cl^-$ uptake [336]. Saline conditions reduce the regulation of nitrate uptake, metabolism, and utilization by plant species. The form of nitrogen applied affects the uptake of other nutrients like $Na^+$, $Ca^{2+}$, and $K^+$, and thus affects the plant's ability to tolerate saline conditions. Therefore, under saline conditions, a nitrogen fertilizer regime should be managed specifically considering the interactions between Na, $NO_3$, $NH_4$, $Cl^-$, etc. [337]. Potassium nitrate is a salt stress alleviating agent for melons [31].

Higher salt concentration in soil solutions interferes with the transport of nitrate in the shoots. For that reason, nitrate reductase, glutamine synthetase, and nitrite reductase enzymes all have reduced activity under stress conditions [246,338]. Nitrate reductase is responsible for the reduction of nitrate to ammonia. Nitrate reductase activity is generally less susceptible to salinity than nitrite reductase activity [339]. Increasing salinity levels decreased dry weight and protein contents of the leaves and roots of tomato seedlings. High Na and chloride ion levels suppressed the uptake of K and nitrate [340].

Nitrogen-containing compounds (NCC) accumulate in plants as a response to higher saline conditions. These compounds include amino acids, amides, quaternary ammonium compounds, and polyamines. Their reported functions are osmoprotection, osmotic adjustment, ROS scavenging, nitrogen provision, and maintenance of pH. Plant nitrogen metabolism can be regulated by phytohormone like cytokinin [29,341].

Phosphorus (P) is a prime constituent in nucleic acids, phospholipids, phosphoproteins, nucleotides, and ATP. Phosphorus uptake is reduced in plants growing under saline conditions. The availability of P is reduced due to strong ionic effects and low solubility of calcium phosphate minerals [302]. Navarro et al. [342] reported a decline in phosphorus mobility stored in vacuoles in salt-stressed melon plants. Plant growth promoting bacteria increased uptake of P and K in tomato plants grown under salt stress. P uptake may enhance the survivability of young plants in salt stress conditions [343–346].

## 10. Conclusions

Plants respond to salinity stress at physiological, cellular, genetic, and metabolic levels. Previous research demonstrates that among plant responses to salinity, mechanisms that control ion uptake, transport, and balance, as well as water potential, photosynthesis, cell division, osmotic adjustment, enzymatic activities, polyamine regulation, stress signaling, and regulation of root apoplastic barriers play critical roles in plant tolerance to salinity. In order to manage salt stress, significant work has been done on calcium biosensors to understand cation-sensing processes, ABA-dependent phosphorylation, changes in cell wall components, auxin and ABA associated modulations in root architecture, $Na^+$ exclusion mechanisms, signaling of phytohormones in roots and guard cells, and organ-specific expression of sodium transporter gene (*HKT1*). There is a need to integrate information from genomic, transcriptomic, proteomic, and metabolomics studies, as a collaborative approach for determining key pathways controlling salinity tolerance at the whole-plant level. Further studies on tissue-specific Na-sensing processes, association of Na and K biosensors with $Na^+/K^+$ homeostasis, ABA and other phytohormone signaling pathways, and interaction between phytohormones and ion transporters are required to illuminate the details of inter- and intracellular molecular interactions that are involved in plant stress tolerance. Further research in these areas will be helpful for mitigating salinity damage in commercially important crops.

**Author Contributions:** Conceived of the presented idea, M.A.S., A.S. and N.K.; developed the theory and performed the computations, M.A.S., N.K., R.M.B. and L.R.; writing—original draft preparation, M.A.S., A.S., N.K., S.A., C.G., N.M., W.N. and F.G.-S.; writing—review and editing, M.A.S., A.S., N.K. and S.A.; supervision, M.A.S. and N.K.; verified the numerical results, R.M.B., S.A., L.R., C.G., N.M., W.N. and F.G.-S. All authors have read and agreed to the published version of the manuscript.

**Funding:** This research received no external funding.

**Conflicts of Interest:** The authors confirm that there is no known conflict of interest associated with this work.

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
