# Peer review of "Insights into the Physiological and Biochemical Impacts of Salt Stress on Plant Growth and Development"

_agronomy, doi:10.3390/agronomy10070938_

Round 1

Reviewer 1 Report

The manuscript is constructed quite well, it contains most of the most necessary paragraphs, which comprehensively discuss the problem of salinity stress. However, the information in individual paragraphs are described rather chaotically, very often too generally. Subject related to salinity stress and the reaction of plants to this stress are very well studied, which is why in my opinion the authors should examine this subject more thoroughly.

Abstract

Abstract need correction, the information in this paragraph should be more consistent.

Key words

Should be listed alphabetically.

Introduction

Line 50 – Flower (1997) – should be changed according to the citation rules.

Line 141-142 - Chartzoulakis and Klapaki, (2000), Kaya et al., (2007) and Giuffrida et al., (2014) -should be changed according to the citation rules. These rules should be implemented throughout manuscript. Please check if these unnumbered citation are in References, because I didn't find them there.

Line 187 – 188 – “Vicia faba, Glycine max, peas and Casuarina glauca…” If several plant species are mentioned, the names should be unify, it will be better to use Latin names of plants.

Line 199-200 – “This energy is then utilized in various basic plant physiological activities like ionic homeostasis in cells, cell division, and elongation.” It is not true. The reader may mistakenly understand that the process of photosynthesis produces energy used, among other things, to transport substances within the plant, and this is not true.

Line 206-207 – I don’t understand the sentence. How is it related to the previous part of the paragraph?

Line 243-252 – This part of paragraph contains information that has nothing to do with each other. Each of these issues should be clarified and described more broadly.

Salinity and water relations

In this paragraph Authors should added some information about the most commonly measured indicators of water relations in plants and how they cope with salinity stress.

Line 301 – what does ‘ROX species’ mean?

Line 337 – “Ethephon, when used with sodium chloride in spinach improved the proline contents [165]” This sentence doesn’t match to this paragraph.

Line 467-488 – This paragraph is messy, there are information about non-enzymatic free radical scavengers, about polyamines, although the chapter was supposed to contain information about enzymes. I suggest moving this information to the appropriate paragraphs.

Line 492-493 - Plant hormones include only 5 groups of compounds (auxins, cytokinins, gibberellins, abscisic acid and ethylene), the rest are classified as growth regulators.

Line 520-550 – Information about brassnosteroids should be moved to different paragraph. They are not classify as plant hormones.

Line 552 – “Salinity exerted progressive depletion of carbohydrates in leaves and roots of citrus” There are many sentences like this in all manuscript. Authors write short sentence with any additional information, what kind of salinity stress was it? How did the amount of carbohydrates change?

Line 566 – 585 – It is very important Paragraph, but Authors doesn’t give any examples how specific species react to salinity.

Line 764 – Salinity and signal transduction – This paragraph should be more precisely described.

Conclusion

This paragraph should include the latest research on salinity stress. Authors should added this information.

References

There are a lot of mistakes in references. Authors should read ‘reference style’ in guide for authors.

Author Response

The manuscript is constructed quite well, it contains most of the most necessary paragraphs, which comprehensively discuss the problem of salinity stress. However, the information in individual paragraphs are described rather chaotically, very often too generally. Subject related to salinity stress and the reaction of plants to this stress are very well studied, which is why in my opinion the authors should examine this subject more thoroughly.

Response: We would like to thank the Reviewer for his/her evaluation and for the constructive comments and suggestions that have helped us improve the quality of the manuscript. We have revised the manuscript following your suggestions and comments to improve its quality. We hope that our revised version would now meet your expectation. Please see below our responses to your comments/suggestions.

Abstract

Abstract need correction, the information in this paragraph should be more consistent.

Response: Abstract has been rewritten to make it more consistent.

Key words

Should be listed alphabetically.

Response: All the keywords are arranged in alphabetical order.

Introduction

Line 50 – Flower (1997) – should be changed according to the citation rules.

Line 141-142 - Chartzoulakis and Klapaki, (2000), Kaya et al., (2007) and Giuffrida et al., (2014) -should be changed according to the citation rules. These rules should be implemented throughout manuscript. Please check if these unnumbered citation are in References, because I didn't find them there.

Response: The MS has been checked thoroughly for all such unnumbered citations and are arranged according to journal format. The missing references are added into the reference list.

Line 187 – 188 – “Vicia faba, Glycine max, peas and Casuarina glauca…” If several plant species are mentioned, the names should be unify, it will be better to use Latin names of plants.

Response: This has been corrected.

Line 199-200 – “This energy is then utilized in various basic plant physiological activities like ionic homeostasis in cells, cell division, and elongation.” It is not true. The reader may mistakenly understand that the process of photosynthesis produces energy used, among other things, to transport substances within the plant, and this is not true.

Response: I am thankful to the reviewer for pointing out this mistake. I have deleted these lines from the MS.

Line 206-207 – I don’t understand the sentence. How is it related to the previous part of the paragraph?

Response: I revised the sentence to provide clarity.

Line 243-252 – This part of paragraph contains information that has nothing to do with each other. Each of these issues should be clarified and described more broadly.

Response: I revised the entire sub-section to provide more clarity.

Salinity and water relations

In this paragraph Authors should added some information about the most commonly measured indicators of water relations in plants and how they cope with salinity stress.

Response: I added a new citation (Salt Tolerance Mechanisms of Plants, 2020 by Eva van Zelm,∗ Yanxia Zhang,∗ and Christa Testerink) to highlight examples of commonly measured parameters that indicate the relationship between water relations and salt stress.

Line 301 – what does ‘ROX species’ mean?

Response: The term ROX was misspelled. It has been corrected and explained in the MS as reactive oxygen species (ROS).

Line 337 – “Ethephon, when used with sodium chloride in spinach improved the proline contents [165]” This sentence doesn’t match to this paragraph.

Response:  The line has been moved to an appropriate place (Line 317).

Line 467-488 – This paragraph is messy, there are information about non-enzymatic free radical scavengers, about polyamines, although the chapter was supposed to contain information about enzymes. I suggest moving this information to the appropriate paragraphs.

Response: The paragraph has been moved to an appropriate place (Lines 238-249).

Line 492-493 - Plant hormones include only 5 groups of compounds (auxins, cytokinins, gibberellins, abscisic acid and ethylene), the rest are classified as growth regulators.

Response: This correction is made to the MS.

Line 520-550 – Information about brassnosteroids should be moved to different paragraph. They are not classify as plant hormones.

Response: Brassinosteroids are kept under a separate subsection ‘’Salinity and Growth Regulators’’.

Line 552 – “Salinity exerted progressive depletion of carbohydrates in leaves and roots of citrus” There are many sentences like this in all manuscript. Authors write short sentence with any additional information, what kind of salinity stress was it? How did the amount of carbohydrates change?

Response: The sentences have been reworded to clear the meaning and readability.

Line 566 – 585 – It is very important Paragraph, but Authors doesn’t give any examples how specific species react to salinity.

Response: Suggestion is addressed accordingly.

Line 764 – Salinity and signal transduction – This paragraph should be more precisely described

Response: This sub-section is removed and these information are moved to an appropriate paragraph (Lines 82-93).

Conclusion

This paragraph should include the latest research on salinity stress. Authors should added this information.

Response: Conclusion is modified according to the reviewer’s comments.

References

There are a lot of mistakes in references. Authors should read ‘reference style’ in guide for authors.

Response: The reference list is modified according to journal format.

Reviewer 2 Report

   The author explores the tolerance mechanism of crops under high salt stress from a wide angle, and uses cell membrane stability as an indication to detect the degree of injury of plants under high salinity.  The article explains the reasons for halophytes, the plants can survive in high salinity, from the aspects of tissue structure, osmotic adjustment, antioxidant enzymes, polyamines, etc., Moreover, the article cited many references to confirm the above content.

     The content of the article is rigorous and substantial, and it is recommended to publish it.

Author Response

The author explores the tolerance mechanism of crops under high salt stress from a wide angle, and uses cell membrane stability as an indication to detect the degree of injury of plants under high salinity.  The article explains the reasons for halophytes, the plants can survive in high salinity, from the aspects of tissue structure, osmotic adjustment, antioxidant enzymes, polyamines, etc., Moreover, the article cited many references to confirm the above content.

     The content of the article is rigorous and substantial, and it is recommended to publish it.

Response: I am thankful to the reviewer for accepting our paper for publication.

Round 2

Reviewer 1 Report

Thank you very much for accepting and correcting all my comments. At the same time, please read the work carefully and correct any inaccuracies. Below are some remarks that I noticed when re-reading the manuscript.

Line 34 – should be phytohormones instead of phytohormone.

Line 173 – 174 – the sentence have wrong construction.

Line 192 – et al. sometimes is Italic, sometimes not. I am asking for harmonization throughout the text.

Line 369 and 374 – is still ROX instead of ROS.

Line 383 – should be spermine instead of spermines.

Line 406- 407 – The plant family names should be italic.

Author Response

Thank you very much for accepting and correcting all my comments. At the same time, please read the work carefully and correct any inaccuracies. Below are some remarks that I noticed when re-reading the manuscript.

Line 34 – should be phytohormones instead of phytohormone.

Response: Correction is made.

Line 173 – 174 – the sentence have wrong construction.

Response: These lines are rephrased.

Line 192 – et al. sometimes is Italic, sometimes not. I am asking for harmonization throughout the text.

Response: Correction is made throughout the MS.

Line 369 and 374 – is still ROX instead of ROS.

Response: We are thankful to the reviewer for pointing out this mistake. It has been corrected.

Line 383 – should be spermine instead of spermines.

Response: Correction is made.

Line 406- 407 – The plant family names should be italic.

Response: Family names are italicized in the MS.